# Expanding Molecular Coverage in Mass Spectrometry Imaging of Microbial Systems Using Metal-Assisted Laser Desorption/Ionization

Jessica K. Lukowski,[a] Arunima Bhattacharjee,[a] Sarah M. Yannarell,[b] Kaitlyn Schwarz,[a] Leslie M. Shor,[c] Elizabeth A. Shank,[d] Christopher R. Anderton[a]

aEnvironmental Molecular Sciences Division, Pacific Northwest National Laboratory, Richland, Washington, USA
bUniversity of North Carolina at Chapel Hill, Chapel Hill, North Carolina, USA
cUniversity of Connecticut, Storrs, Connecticut, USA
dUniversity of Massachusetts Medical School, Worcester, Massachusetts, USA

**ABSTRACT** Mass spectrometry imaging (MSI) is becoming an increasingly popular analytical technique to investigate microbial systems. However, differences in the ionization efficiencies of distinct MSI methods lead to biases in terms of what types and classes of molecules can be detected. Here, we sought to increase the molecular coverage of microbial colonies by employing metal-assisted laser desorption/ionization (MetA-LDI) MSI, and we compared our results to more commonly utilized matrix-assisted laser desorption/ionization MALDI MSI. We found substantial (~67%) overlap in the molecules detected in our analysis of *Bacillus subtilis* colony biofilms using both methods, but each ionization technique did lead to the identification of a unique subset of molecular species. MetA-LDI MSI tended to identify more small molecules and neutral lipids, whereas MALDI MSI more readily detected other lipids and surfactin species. Putative annotations were made using METASPACE, Metlin, and the BsubCyc database. These annotations were then confirmed from analyses of replicate bacterial colonies using liquid extraction surface analysis tandem mass spectrometry. Additionally, we analyzed *B. subtilis* biofilms in a polymer-based emulated soil micromodel using MetA-LDI MSI to better understand bacterial processes and metabolism in a native, soil-like environment. We were able to detect different molecular signatures within the micropore regions of the micromodel. We also show that MetA-LDI MSI can be used to analyze microbial biofilms from electrically insulating material. Overall, this study expands the molecular universe of microbial metabolism that can be visualized by MSI.

**IMPORTANCE** Matrix-assisted laser desorption/ionization mass spectrometry imaging is becoming an important technique to investigate molecular processes within microbial colonies and microbiomes under different environmental conditions. However, this method is limited in terms of the types and classes of molecules that can be detected. In this study, we utilized metal-assisted laser desorption/ionization mass spectrometry imaging, which expanded the range of molecules that could be imaged from microbial samples. One advantage of this technique is that the addition of a metal helps facilitate ionization from electrically nonconductive substrates, which allows for the investigation of biofilms grown in polymer-based devices, like soil-emulating micromodels.

**KEYWORDS** MetA-LDI, MALDI, *Bacillus subtilis*, microfluidics, soil microbiome, biofilms

Address correspondence to Christopher R. Anderton, Christopher.Anderton@pnnl.gov.

Mass spectrometry imaging (MSI) is an analytical technique that can spatially detect and identify molecules within cells, tissues, or organisms. MSI methods employ an analytical probe (e.g., ion beam, laser, or solvent junction) capable of *in situ*

endogenous chemical desorption and/or ionization (1–4). Of these methods, matrix-assisted laser desorption/ionization mass spectrometry imaging (MALDI MSI) was recently surveyed to be the most popular MSI method utilized for molecular imaging (5). MALDI MSI is a soft ionization technique that allows for a wide range of intact molecules to simultaneously be analyzed with high biomolecular specificity. Over the last decade, numerous reports have demonstrated the usefulness of MALDI MSI for studying microbial systems (6–9). The chemical biomolecular specificity (i.e., the general classes of biomolecules detectable) can be tailored, in part, by the choice of matrix used for MALDI MSI analysis (10). For example, in positive-ion-mode analysis, sinapinic acid is typically used for intact protein measurements, while 2,5-dihydroxybenzoic acid (DHB) and $\alpha$-cyano-4-hydroxycinnamic acid have been commonly used for both lipid analysis and small molecule analysis (11).

An alternative to MALDI is metal-assisted laser desorption/ionization (MetA-LDI). In this technique, metals are sputter coated or sprayed onto the sample's surface at various thicknesses to aid in the ionization of endogenous molecules (12), much like that of the organic acid matrices used in MALDI. Some of the advantages of using metals for MSI analysis include higher laser absorptivity in the UV range, nonvolatility (i.e., stable under high vacuum), no matrix interference in the low-mass range ($m/z$ <500), and more homogenous film formation (i.e., fewer "hot spots" and analyte delocalization) (11, 13, 14). Predominately, gold and silver are used for metal-assisted ionization due to their ability to increase the ionization efficiency of cholesterol, fatty acids, and other olefin-related compounds (10, 14, 15). Previous work utilizing MetA-LDI MSI shows that this technique can increase the number of neutral molecules identified, therefore complementing MALDI MSI by increasing the overall molecular coverage that can be detected by LDI MS instrumentation. For example, Dufresne et al. investigated the effects of sodium salts and Au sputter coating on the surface of mammalian tissue sections for LDI analysis (16). They found that depositing both sodium salts and a 28-nm thickness of Au on their sample increased the number of triglycerides detected by 350% versus MALDI MSI utilizing DHB as a matrix (16). Furthermore, Dunham et al. utilized MetA-LDI and metal-enhanced $C_{60}$-secondary ion mass spectrometry to image bacterial biofilms, where they observed excellent spectral purity and image quality of small molecules, specifically rhamnolipids and 2-alkyl-quinolones (17). These two papers highlight the benefits of using metals for MSI and show how MetA-LDI can be used to expand the molecular coverage of the sample being interrogated.

One area of research that has yet to exploit the benefits of MetA-LDI MS is the molecular imaging of microbial communities. Here, we used *Bacillus subtilis*, a Gram-positive bacterium commonly found in soil, as a model species to compare MetA-LDI MSI and MALDI MSI of bacterial communities. *B. subtilis* can form biofilms, or communities of cells encased in an extracellular matrix, that are known for tolerating harsh environmental conditions (18, 19). Additionally, we analyzed *B. subtilis* biofilms in an emulated soil micromodel using MetA-LDI MSI (20). The metal coating applied in the MetA-LDI approach renders electrically insulating materials conductive and available for MSI analysis without the use of an organic acid matrix to aid in the ionization. To demonstrate this, we investigated *B. subtilis* grown in UV-curable polymer soil micromodels that simulate the physical characteristics of soil, such as porosity and aggregate sizes (see Fig. S1 in the supplemental material) (21). This permitted us to spatially uncover molecular processes of *B. subtilis* in a setting more similar to soil than agar plates, something that has not previously been done.

## RESULTS AND DISCUSSION

Previous work on mammalian and plant tissues has shown MALDI and MetA-LDI approaches ionize some similar molecules, but that each technique can ionize a unique subset of molecules as well (14, 17). Here, we first optimized sample preparation with regard to matrix deposition and metal sputter coating for each ionization technique to expand and maximize the molecular coverage of metabolites from microbial colonies.

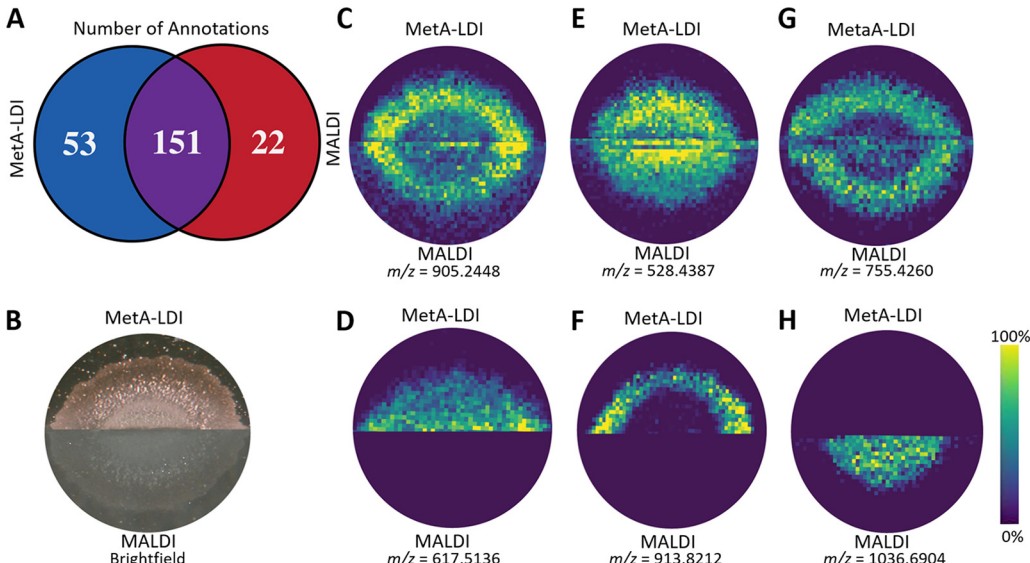

**FIG 1** Comparison of MetA-LDI MSI and MALDI MSI of a *B. subtilis* colony grown on agar. (A) Venn diagram illustrating the overlap of annotations found in the two MSI techniques. (B) Bright-field image of the colony analyzed by both techniques after gold sputter coating (MetA-LDI) and DHB matrix deposition (MALDI). (C to H) Representative ion images of (C) corynebactin at $m/z$ 905.2448 $[M+Na]^+$, (D) DG (36:4) at $m/z$ 617.5136 $[M+H]^+$, (E) Cer (32:3) at $m/z$ 528.4387 $[M+Na]^+$, (F) TG (56:3) at $m/z$ 913.8212 $[M+H]^+$, (G) PG (32:3) at $m/z$ 755.4260 $[M+K]^+$, and (H) surfactin C at $m/z$ 1,036.6904 $[M+H]^+$ of the colony analyzed.

For MALDI MSI, a DHB matrix (40 mg/ml in 70% MeOH) was applied for 8, 10, 12, or 14 cycles to *B. subtilis* colonies at 65°C using an automated sprayer (22). We observed that the maximum number of molecular species was detected when 10 passes of DHB matrix were applied to the microbial colony. This was evaluated by obtaining the combined maximum number of annotations from METASPACE (23), Metlin (24), and the *B. subtilis* database (25). The same optimization process was then performed to determine the optimal thickness of Au to be applied to the surface of the *B. subtilis* colonies for MetA-LDI MSI analysis. Microbial colonies were sputter coated with thicknesses of 5, 7, 8, 10, 15, or 20 nm of Au and analyzed with MetA-LDI-MSI to determine the maximum number of annotations using the same metabolite databases as in our MALDI MSI analysis. As such, we found an optimum Au film thickness of 8 nm based on producing the maximum number of annotations. The number of annotations found under all conditions is summarized in Table S1 in the supplemental material.

With the sample preparation steps optimized, we then performed a direct comparison between the two mass spectrometry ionization techniques. For this, *B. subtilis* colonies were grown overnight and cut in half, and each half was placed on separate slides: one for MetA-LDI MSI analysis and one for MALDI MSI analysis. Comparison of the microbial colony halves in triplicate by MetA-LDI MSI and MALDI MSI demonstrated a substantial (~67%) overlap between the annotations from the respective analysis data sets (Fig. 1A). While there was a significant overlap in annotations, each ionization technique did identify a unique subset of molecular species. MetA-LDI MSI tended to identify more small molecules and neutral lipids, while MALDI MSI better identified other lipids, such as phosphatidylcholines (PCs) and phosphatidylethanolamines (PEs), and surfactin species (see Table S2 in the supplemental material). Figure 1C to H illustrate representative ion images of several molecular species found in each dataset. Figure 1C shows the molecular signal obtained from corynebactin, also known as bacillibactin, which can be detected using both MetA-LDI MSI and MALDI MSI. This species was visualized in the outermost parts of the bacterial colony. Also detected in both data sets was ceramide (Cer) (32:3) (Fig. 1E). This species is present throughout the colony, but a higher intensity of this species could be visualized at the center of the colony using MALDI MSI. Figure 1G shows phosphatidylglycerol (PG) (34:3), which has a

**TABLE 1** LESA MS/MS confirmed species of *B. subtilis* colony grown on agar

| Molecular species | Parent ion [M + H]$^+$ | No. of matching fragments identified by MetFrag | Detected in data set from: | |
| --- | --- | --- | --- | --- |
| | | | MetA-LDI | MALDI |
| DG (36:4) | 617.5136 | 8/15 | X | |
| TG (56:3) | 913.8212 | 9/15 | X | |
| Surfactin C | 1,036.6904 | 10/15 | | X |
| Phallacidin | 847.3225 | 7/15 | | X |
| Cer (32:3) | 506.4513 | 9/15 | X | X |
| PE (31:3) | 672.4599 | 8/15 | X | X |
| Corynebactin | 883.2579 | 10/15 | X | X |

similar distribution to corynebactin appearing toward the outside the colony. Diacylglycerol (DG) (36:4) and triacylglycerol (TG) (56:3) (Fig. 1D and F, respectively) were exclusively found in the MetA-LDI MSI data set, while surfactin C (Fig. 1H) was solely found in the MALDI MSI analysis.

Overall, our findings are consistent with previous experiments. For example, Dufresne et al. sputter coated Ag onto thin tissue sections and found different lipid classes were able to be characterized by LDI MSI than with MALDI MSI (12). Specifically, small fatty acids and triacylglycerides could be identified in their samples, providing new biological insights into disease pathways (12). Additionally, Hansen et al. performed a systematic study comparing six different sputter-coated metals on maize root and seed. They found that Ag, Au, and Pt were much more efficient than other transition metals in the analysis of small metabolites, triacylglycerols, and diacylglycerols (14). These two studies complement our findings, as we saw a significant increase in the number of triacylglycerols and diacylglycerols in our MetA-LDI MSI data set in comparison to the MALDI MSI data set. Additionally, the relative intensity for the species detected in MetA-LDI in comparison to MALDI tended to be slightly higher, as seen by the increased intensity of lysophosphatidylcholine (PC) (24:0), for example (see Fig. S2 in the supplemental material).

To verify the putative molecular annotations found in the MetA-LDI MSI and MALDI MSI experiments, liquid extraction surface analysis tandem mass spectrometry (LESA MS/MS) was performed on replicate *B. subtilis* colonies. Briefly, LESA is an ambient mass spectrometry technique that allows for the direct analysis of analytes from a surface (26). In this experiment, 70% MeOH was used as the extraction solvent for the analysis of molecular species from the bacterial colony (the same solvent used for matrix addition for the MALDI MSI experiments). Liquid extractions were performed at the colony edge and center to ensure detection of a wide range of molecular species that were representative of the entire colony. Table 1 shows a short list of notable molecular species that were found uniquely in the MetA-LDI MSI or MALDI MSI analysis or that were detected using both methods, whose identities were verified by LESA MS/MS. A more detailed list of the molecular species verified by LESA MS/MS can be found in Table S3 in the supplemental material. MetA-LDI MSI and MALDI MSI have slight ionization differences and can suffer from in-source fragmentation, so LESA MS/MS analysis of the replicate samples was able to confirm the presence and identity of species that were detected in each of these approaches. Overall, confirmation of these molecular identifications shows that we are indeed expanding the molecular coverage of microbial species through use of MetA-LDI MSI, and not just observing more in-source fragmentation of endogenous species, for example.

One major limitation to analyzing soil bacterial processes in colonies grown on nutrient agar is that it does not accurately represent the complexity and heterogeneity of the soil matrix (7). Therefore, we also analyzed *B. subtilis* biofilms grown in emulated soil micromodels to examine how *B. subtilis* behaves in a more soil-like environment. These soil micromodels are designed to emulate the particle size distributions, aggregate

sizes, and pore size distributions of a sandy loam soil and have three identical microstructured channels that provide built-in technical replicates for bacterial biofilm growth (Fig. S1) (20, 21). Previously, these micromodels were used to measure the water retention properties of a soil bacterium's extracellular polymeric substance (21) and dynamic interactions of bacteria (21) and fungi (27) within structured microenvironments, while similar systems have been used to study transport of bacteria in flowing micromodel systems (28) and particle transport by soil protists in quiescent micromodel systems (29), in each case revealing phenomena that only occur within an appropriately scaled and suitably physiochemically complex soil pore microenvironment. The soil micromodels are fabricated from polymers that are electrically insulating (28). By coating the sample with Au, for MetA-LDI MSI analysis, this rendered these devices electrically conductive and permitted us to spatially visualize a portion of the metabolome of *B. subtilis* in a soil-like environment.

To illustrate that these micromodels were compatible with MetA-LDI MSI analysis, a standard lipid (1-oleoyl-2-palmitoyl-sn-glycero-3-phosphocholine) was flowed through the empty channels and lyophilized. The microfluidic device was then sputter coated with Au and analyzed with MetA-LDI MSI. As such, an image of the standard lipid in the soil macropore spaces of the micromodel could be visualized (see Fig. S3 in the supplemental material). During this experiment, we noticed some space charge effects that attenuated the signal in certain regions of the channel when analyzed as a whole (see Fig. S4 in the supplemental material). The top of the channel could be normally visualized, but as the laser raster moved down the channel, fewer ions were visualized. However, space charging did not affect the analysis when the channel was spilt into top, bottom, and middle analysis regions versus imaging continuously from top to bottom. In this way, the parameters for direct MetA-LDI MSI on the micromodel surface were optimized for maximum signal-to-noise ratio.

*B. subtilis* cells were inoculated into the microfluidic device and grown until microbial biofilms were detected using confocal microscopy. Specifically, the *B. subtilis* strain used in this study contained a fluorescent transcriptional reporter for biofilm matrix gene expression ($P_{tapA}$-YPet [YPet is a derivative of the yellow fluorescent protein]) that could be detected via high-resolution fluorescence microscopy. After enough cells were expressing the biofilm matrix genes, as verified through fluorescence microscopy (Fig. 2A), the microfluidic devices were lyophilized and sputter coated with 8 nm of Au for MetA-LDI analysis. Figure 2 shows the ion images of four (of 176) different species that were detected within the microfluidic channel. The ion images show that after lyophilization, the lipid species seem to get pushed to the sides and aggregate around the smaller micropore structures. The feature outlined in white in Fig. 2 shows where an air bubble was present in the channel, and it can be clearly seen in the ion images that no lipid species were present in this area. The feature outlined in red in Fig. 2 highlights a micropore aggregate feature within the channel where lipid species could be identified around outside the micropore structures. Moreover, DG (36:3) was detected in the MetA-LDI MSI colony data set, but was not present in the MALDI MSI data set (Fig. 2D). This shows the robustness of this technique that species identified using MetA-LDI MSI in bacterial colonies can also be identified on the microfluidic surface. The higher relative intensity at the edges of the micropore region could result from the use of a 50-$\mu$m spatial resolution which is significantly larger the smallest micropore regions (5 to 10 $\mu$m). In the future, we anticipate that using a finer spatial resolution will enable us to further tease apart the spatial distributions of lipid species that are specific to the soil micropore environment. Overall, these data show that the technique of conducting MetA-LDI MSI directly on the micromodel surface is comparable to agar-based microbial colony analysis. In this way, we can leverage the reduced-complexity structure of soil micromodels to obtain spatial metabolomic data from bacterial communities in soil-like environments at microscale resolution.

This research highlights how MetA-LDI MSI and confocal microscopy can be used in a multimodal fashion to better understand microbial community dynamics. Using the model

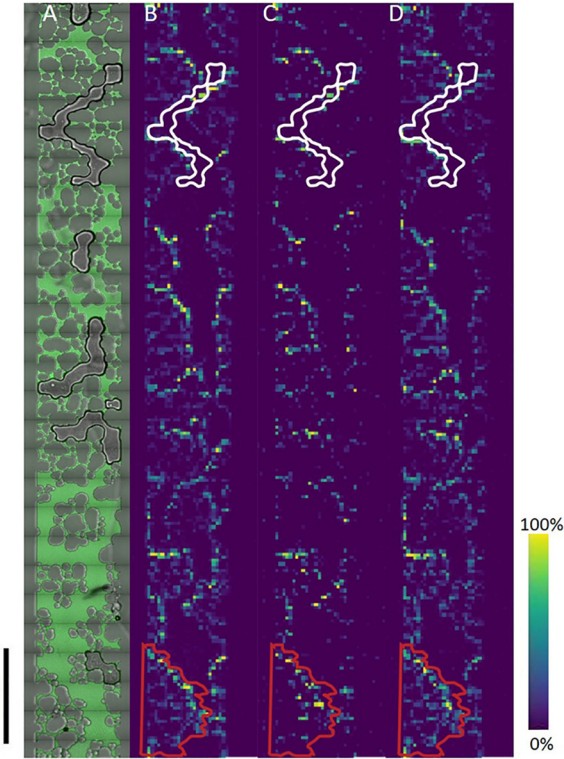

**FIG 2** MetA-LDI MSI of *B. subtilis* grown in the emulated soil micromodel. (A) *B. subtilis* that has formed a biofilm is visualized by fluorescence microscopy prior to lyophilization and gold sputter coating and MetA-LDI MSI analysis. (B) PG (32:0) at *m/z* 723.5171 $[M+H]^+$, (C) PE (34:2) at *m/z* 716.5225 $[M+H]^+$, and (D) DG (36:3) at *m/z* 619.5296 $[M+H]^+$ were able to be visualized throughout the microchannel. The feature outlined in white highlights an area in which an air bubble was present in the microchannel, so minimal lipid signal from *B. subtilis* was detected in the fluorescence microscopy as well as the MetA-LDI MSI analysis. The feature outlined in a red rectangle highlights an aggregate structure within the micromodel, and Meta-LDI MSI analysis revealed lipids could be clearly detected around the perimeter of this feature.

soil bacterium *B. subtilis*, we visualized microbial biofilm formation and various interactions by use of fluorescent reporters and then utilized spatial metabolomics to gain direct insights about these interactions. We were able to demonstrate key differences in the types of molecular species that could be identified when analyzing these bacterial communities with MetA-LDI MSI and MALDI MSI. Our results demonstrate that MetA-LDI MSI of these systems can expand the types of molecules that can be detected compared to using MALDI MSI alone. Specifically, we could measure more small molecules and neutral lipids in *B. subtilis* using MetA-LDI MSI, which were not detected in our MALDI MSI analysis. Furthermore, by using the MetA-LDI MSI approach, we could obtain spatial metabolomics of *B. subtilis* communities grown in emulated soil micromodels. Combined, this work established the usefulness of MetA-LDI MSI for molecular imaging of microbes and their environments, adding to the arsenal of tools our community can employ for directly visualizing microbial metabolic processes and expanding our ability to understand the mechanisms of interkingdom interactions in the soil.

## MATERIALS AND METHODS

**Bacterial strain and culture conditions.** *Bacillus subtilis* NCIB 3610 *amyE*::P*tapA*-*Ypet* strain (ES2107) was used for all experiments. The strain was cultured on lysogeny broth (LB)-agar plates prepared from Miller LB (Acros Organics, Fair Lawn, NJ) and 1.5% agar powder (Molecular Genetics, Fisher Scientific, Waltham, MA). LB agar plates were streaked with *B. subtilis* ES2107 from frozen stocks and incubated at 30°C for 15 h.

*B. subtilis* biofilms were grown on MSgg, a *B. subtilis* biofilm-promoting medium (5 mM potassium phosphate [pH 7]) (Fisher Scientific, Waltham, MA), 100 mM MOPS (morpholinepropanesulfonic acid [pH 7]) (Sigma-Aldrich, St. Louis, MO), 2 mM $MgCl_2$ (Fisher Scientific, Waltham, MA), 700 μM $CaCl_2$ (Alfa Aesar, Haverhill, MA), 50 μM $MnCl_2$ (Fisher Scientific, Waltham, MA), 50 μM $FeCl_3$ (Sigma-Aldrich, St. Louis, MO),

$1\,\mu M$ ZnCl$_2$ (Sigma-Aldrich, St. Louis, MO), $2\,\mu M$ thiamine hydrochloride (Fisher Scientific, Waltham, MA), 0.5% glycerol (Fisher Scientific, Waltham, MA), and 0.5% glutamate (Sigma-Aldrich, St. Louis, MO). For MSgg agar plates, 1.5% agar (BD) was added to the medium, and approximately 5-ml plates were poured.

**MALDI and MetA-LDI MSI sample preparation and data acquisition.** *B. subtilis* ES2107 colonies for MALDI- and MetA-LDI MSI were prepared by inoculating a single colony from routine overnight growth on LB agar in 1 ml MSgg medium. Then MSgg agar plates were spotted with $0.5\,\mu l$ of inoculated MSgg medium and incubated at 30°C for 15 h. For analysis, the colony and agar surrounding the colony were cut using a disposable scalpel, removed from the plate, and mounted on indium tin oxide (ITO)-coated glass slides (Bruker, Billerica, MA). Each colony was dried overnight in a biological safety cabinet prior to further preparation for MSI.

For MALDI MSI, samples were coated using an M5 TM-Sprayer (HTX Technologies). A 40-mg/ml concentration of DHB in 70% MeOH-H$_2$O was applied using the following spraying parameters: 65°C nozzle temperature, flow rate of 0.1 ml/min, 12 passes, N$_2$ pressure of 10 lb/in$^2$, track spacing of 3 mm, and 40-mm distance between the nozzle and sample. For MetA-LDI MSI, the dried bacterial colonies on ITO-coated glass slides were coated with thin films of Au using a 208 HR Cressington sputter coating system (Cressington Scientific Instruments, UK). Several thicknesses of Au, such as 5, 7, 8, 9, 10, 15, and 20 nm, were tested using MetA-LDI for optimal mass spectrometry signal (Table S1).

MSI was performed using a 15 Tesla Fourier transform ion cyclotron resonance (FTICR) mass spectrometer (Bruker Daltonics) equipped with a SmartBeam II laser source (355 nm, 2 kHz) in positive-ion mode using 200 shots/pixel at 35% laser power and an 85-$\mu m$ pitch between pixels. FTICR MS was externally calibrated using an Agilent ESI (electrospray ionization) Low Concentration Tuning mix and operated to collect *m/z* 300 to 1,300 using a 577-ms transient that translated to a mass resolving power of $\sim$170,000 at 400 *m/z*. The ion transfer and analyzer parameters remained constant based on optimizing the system using the ESI source to allow maximal transmission and detections of signals of *m/z* 600 to 900. MSI data were acquired using FlexImaging v4.1 (Bruker Daltonics).

**MSI data processing.** FTICR MS imaging data were imported into the SCiLS software (SQLite format; Bruker Daltonics) and converted into the imzML format with spectrum restriction using only the *m/z* intervals of the imported peaks. The resulting imzML and ibd files were then uploaded to METASPACE (https://metaspace2020.eu) (23). The data were annotated using a custom *B. subtilis* database Annotated species with observed localization predominantly off-sample were filtered from the list. From METASPACE, CSV files were downloaded for each replicate, and annotations were cross-referenced to create a list consisting of the overlapping annotations. All data sets in METASPACE can be accessed at https://metaspace2020.eu/project/LDI_microbial_communities. Additionally, peak lists were entered into the Metlin (https://metlin.scripps.edu) (24) and BsubCyc (https://bsubcyc.org/) (25) databases to increase the number of molecular species identified. A complete list of the annotations can be found in an Excel document in the supplemental material.

**LESA MS/MS analysis.** *B. subtilis* ES2107 colonies grown under the same protocol as described above and were analyzed by LESA MS/MS using a Triversa Nanomate (Advion) coupled to a Velos Orbitrap mass spectrometer (Thermo Scientific) (30). The LESA sampling conditions were as follows: $8.0\,\mu l$ of 70% MeOH with 0.1% formic acid was aspirated into the pipette tip, then $0.5\,\mu l$ of this volume was dispensed at a height of 0.2 mm above the sample surface. After a 1-s postdispense delay (extraction) time, the volume was reaspirated into the pipette tip and then infused into the MS via nano-electrospray ionization. Each sample extraction was sprayed into the MS for 4 min using a data-dependent tandem MS method, as previously described (30). Briefly, a high-mass-resolution MS scan was collected (*m/z* 100 to 1,300), and then the six most intense ions from the target list were selected for collision-induced dissociation (CID) fragmentation from the first MS scan. This process was repeated, and the targeted ions were then dynamically excluded for the next 30 s. The tandem MS data were compared with the *in silico* fragmentation tool (MetFrag) to confidently identify target molecules (31). A complete list of the fragments and molecules identified can be found in an Excel document in the supplemental material. Three *B. subtilis* colonies were analyzed in positive-ion mode as biological replicates.

**Microfluidic device fabrication.** The emulated soil micromodels were constructed using a combination of the Bosch process (32) and soft lithography techniques (33). A diagram illustrating this process is shown in Fig. S1. Briefly, a positive Si master of the microfluidic channel, previously described (20), was constructed using the Bosch process and functionalized using chlorotrimethylsilane (CTMS; Sigma-Aldrich, St. Louis, MO) by exposure in a desiccator under vacuum overnight. Negative replicas of the Si master were produced from polydimethylsiloxane (PDMS) (Sylgard 184; Dow Corning, Midland, MI) with a prepolymer-to-curing agent ratio of 10:1. After extensive mixing of the prepolymer and curing agent, the mixture was poured on the Si master of the microchannel and placed in a vacuum desiccator for 1 h to eliminate all air bubbles. PDMS castings on masters were then thermally cured in an oven for 3 h at 70°C. After cooling, the negative PDMS mold was gently peeled off the Si substrate. The negative PDMS mold was then treated in oxygen plasma for 30 s in plasma cleaner (PX250; Nordson March, Concord, CA). After this surface treatment, the PDMS negative mold was placed in CTMS environment in a desiccator under vacuum overnight.

In order to produce the final positive replica of the microchannel, glass coverslips were functionalized by placing them in an aminopropyltrimethoxysilane (Sigma-Aldrich, St. Louis, MO) environment in a vacuum desiccator overnight. A layer of OG603 epoxy (Epoxy Technology, Inc., Billerica, MA) was spin coated using a WS-400B-6NPP/LITE coater (Laurell Technologies Corp., North Wales, PA) to the functionalized glass coverslip at a spin speed of 500 rpm for 10 s. Then, the PDMS negative channel was filled with OG603 epoxy and placed under vacuum in a desiccator to ensure the air bubbles were eliminated and the epoxy filled the entire microchannel space. The microchannel surface with epoxy was then placed against the epoxy-coated glass coverslip and pressed to remove excess epoxy. This setup was

then cured under UV radiation (Melody Sussie UV gel nail polish dryer) overnight to solidify. The negative PDMS replica was then peeled off, leaving the epoxy microchannel positive with a glass coverslip backing. This process produced epoxy films that were 5 to 7 $\mu$m thick and required a glass coverslip backing to increase their structural integrity.

**Microfluidic device assembly and *B. subtilis* ES2107 inoculation.** The epoxy microfluidic channels were plasma cleaned and reversibly bonded to a 5-mm PDMS layer with ports at two ends (Fig. S1) for inoculation of *B. subtilis* ES2107. For reversible bonding of PDMS to the epoxy, a 5-mm PDMS layer was created by pouring 40 g of a 20:1 PDMS prepolymer-to-curing agent ratio mixture into a 150- by 15-mm petri dish (VWR, Radnor, PA) and curing it at 70°C for 30 min. PDMS covers for epoxy devices were cut out, and ports were created using a 5-mm hole punch, To create inoculation port covers for this devices, we followed the same 20:1 PDMS ratio recipe, and 20 g of this mixture was poured into a 150-mm-diameter petri dish (VWR, Radnor, PA) and cured at 70°C for 30 min. The cured polymer was cut into 6- and 6-mm squares and plasma cleaned (PX250; Nordson March, Concord, CA) before being used to seal inoculation ports of the micromodels.

A single colony of *B. subtilis* ES2107 from overnight growth on LB agar was inoculated in 5 ml MSgg medium and incubated at 37°C and shaking at 250 rpm to a higher cell density and diluted in MSgg medium to an optical density at 600 nm ($OD_{600}$) of 0.01. The inoculum was added to the input port of the device to fill the entire channel. The microfluidic device was incubated at room temperature in the dark for 3 days prior to imaging with confocal microscopy.

**Confocal imaging of *B. subtilis* ES2107 biofilms in microchannels.** All confocal microscopy images were acquired using a confocal microscope (Zeiss, Oberkochen, Germany) with a W Plan Apochromat 20× objective. The YPet protein was excited with 514-nm laser, and the pinhole was adjusted to 4 Airy units. Fluorescence images were collected between 513 and 620 nm, and bright-field images of the channel were acquired simultaneously. Using the tile scan function in ZEN 2.3 SP1 software (Zeiss), several 425.1- by 425.1-$\mu$m images were stitched over all three 10- by 1-mm microchannels (Fig. S1) to acquire mosaic images of the microfluidic device. Moreover, several 10-$\mu$m sections were acquired along the z direction to capture the entire thickness of the device, and a final image of the maximum-intensity projection of all of the z sections was generated. All confocal microscopy images were analyzed using the ZEN image analysis software (Zeiss).

**Lyophilization and sputter coating of microchannels.** The microfluidic device was placed at −20°C after confocal microscopy imaging for 4 h to facilitate removal of the PDMS top cover without disrupting the biofilm. After freezing the biofilm cells, the top PDMS cover was gently peeled off from the epoxy channel. The epoxy channel with glass coverslip backing was placed at −80°C overnight and lyophilized for 2 h. The lyophilized microchannels and bacterial colonies on ITO-coated glass slides were coated with 8-nm thin films of Au using a 208 HR Cressington sputter coating system (Cressington Scientific Instruments, UK). Microfluidic devices were attached to MALDI target using copper tape to ensure that a conductive surface was present. Microfluidic devices were analyzed by splitting the channel into 3 regions of interest to mitigate space charge effects and then analyzed by the same method listed above (Fig. S4).

## SUPPLEMENTAL MATERIAL

Supplemental material is available online only.
**SUPPLEMENTAL FILE 1**, DOCX file, 1.6 MB.
**SUPPLEMENTAL FILE 2**, XLSX file, 0.04 MB.
**SUPPLEMENTAL FILE 3**, XLSX file, 0.3 MB.

## ACKNOWLEDGMENTS

This article is based upon work supported by the U.S. Department of Energy (DOE), Office of Science (award DE-SC0019012 to E.A.S. and C.R.A.), as well as the Office of Workforce Development for Teachers and Scientists, Office of Science Graduate Student Research (SCGSR) program. the SCGSR program is administered by the Oak Ridge Institute for Science and Education for the DOE under contract DE-SC0014664 (to S.M.Y.) L.M.S. was supported by NSF 1605816. This research was performed using the Environmental Molecular Sciences Laboratory, a DOE Office of Science User Facility sponsored by the Office of Biological and Environmental Research and located at Pacific Northwest National Laboratory (PNNL). PNNL is operated for the DOE by Battelle Memorial Institute under contract DE-AC05-76RLO1830.

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
