## [Reviewer comments · Microbiology Spectrum]

Microbiology Spectrum

Expanding the molecular coverage in mass spectrometry imaging of microbial systems using metal-assisted laser desorption/ionization

Jessica Lukowski, Arunima Bhattacharjee, Sarah Yannarell, Kaitlyn Schwarz, Leslie Shor, Elizabeth Shank, and Christopher Anderton

Corresponding Author(s): Christopher Anderton, Pacific Northwest National Laboratory

Review Timeline:

Submission Date:

June 21, 2021

Accepted:

June 22, 2021

Editor: Jeffrey Gralnick

Reviewer(s): The reviewers have opted to remain anonymous.

Transaction Report:

DOI: <https://doi.org/10.1128/Spectrum.00520-21>

June 22, 2021

Dr. Christopher R. Anderton
Pacific Northwest National Laboratory
PO Box 999
MSIN: K8-98
Richland 99352

Re: Spectrum00520-21 (Expanding the molecular coverage in mass spectrometry imaging of microbial systems using metal-assisted laser desorption/ionization)

Dear Dr. Christopher R. Anderton:

After consultation with another Editor, we both agreed that your responses to the prior round of review were thoughtful and thorough. I am pleased to notify you that your manuscript has been accepted, and I am forwarding it to the ASM Journals Department for publication. You will be notified when your proofs are ready to be viewed.

Sincerely,

Jeffrey Gralnick
Editor, Microbiology Spectrum

Journals Department
Supplemental Figures and Tables: Accept

Annotations Table: Accept

MetFrag ID of detected species: Accept